# Learning with Counterfactual Explanations for Radiology Report Generation

## Abstract

Due to the common content of anatomy, radiology images with their corresponding reports exhibit highly similarity. Such inherent data bias can predispose automatic report generation models to learn entangled and spurious representations resulting in misdiagnostic reports. Moreover, the lack of explainability hinders the acceptance by radiologists in clinical practice. To tackle these, we propose a novel **C**ounter**F**actual **E**xplanations-based framework (CoFE) for radiology report generation. Counterfactual explanations serve as a potent tool for understanding how decisions made by algorithms can be changed by asking "what if" scenarios. By leveraging this concept, CoFE can learn non-spurious visual representations by contrasting the representations between factual and counterfactual images. Specifically, we derive counterfactual images by swapping a patch between positive and negative samples until a predicted diagnosis shift occurs. Here, positive and negative samples are the most semantically similar but have different diagnosis labels. Additionally, CoFE employs a learnable prompt to efficiently fine-tune the pre-trained large language model, encapsulating both factual and counterfactual content to provide a more generalizable prompt representation. Extensive experiments on two benchmarks demonstrate that leveraging the counterfactual explanations enables CoFE to generate semantically coherent and factually complete reports and outperform in terms of language generation and clinical efficacy metrics.

## 1 Introduction

Automatically generating reports can reduce the load on radiologists and potentially increase the accuracy and consistency of interpretations. This is achieved by translating intricate radiology images into semantically coherent and clinically reliable free texts. However, in comparison to generic captioning tasks, Radiology Report Generation (RRG) presents a significant challenge, often yielding unsatisfactory performance when employing direct captioning methods Vinyals et al. (2015); Lu et al. (2017) in the field of radiology. The difficulty arises due to the severe data bias within the limited image-report pair data available, a challenge that has been extensively acknowledged and discussed Liu et al. (2021a); Chen et al. (2020); Li et al. (2023); Tanida et al. (2023); Wang et al. (2023).

Given the shared anatomical content, radiology images tend to display significant similarity to one another, with abnormal or lesioned areas typically occupying minor portions of the images Li et al. (2022c); Voutharoja et al. (2023). This similarity also extends to the accompanying reports, where several sentences often describe normal tissues. However, the clinical usefulness of radiology reports hinges on the accurate depiction of abnormalities. This intrinsic data bias tends to lead models to learn spurious and intertwined visual features, resulting in the generation of inaccurate diagnostic reports. To mitigate data bias, various successful concepts have been proposed by existing methods to enhance learning representations, such as employing contrastive learning Li et al. (2023); Liu et al. (2021b), incorporating medical knowledge Liu et al. (2021a); Yang et al. (2023b), and implementing relational memory Chen et al. (2020) etc.

Recently, Tanida *et al.* Tanida et al. (2023) achieved the state-of-the-art (SOTA) performance by detecting abnormal regions using a pre-trained detector with pseudo labels. They then utilized these features to guide a pre-trained large language model (LLM) in generating reports. Identifying critical regions that cover abnormalities or lesions not only enhances non-spurious visual representation

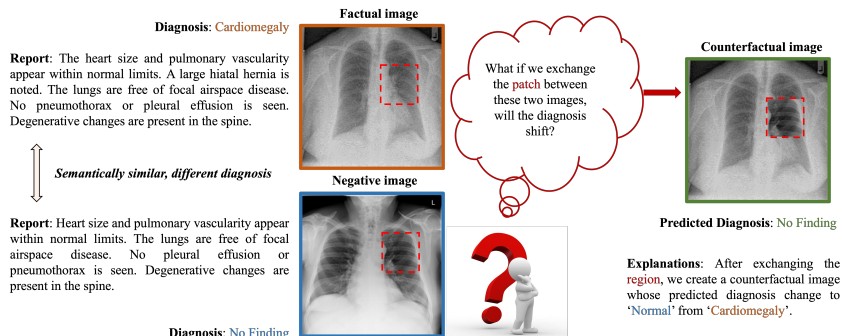

Figure 1: A conceptual overview of our proposed counterfactual explanations is presented. Such CEs help to construct a counterfactual image by iteratively exchanging a patch between factual (positive) and negative images until the predicted diagnosis shift occurs. In this instance, the box in red covering the heart is identified as the critical region that causes the diagnosis shift.

but also improves the explainability of RRG models. Ideally, the method would employ golden annotations to train a lesion detector capable of accurately localizing abnormal regions or lesions. However, existing RRG benchmarks lack such annotations. Relying on weakly supervised signals from pseudo labels Tanida et al. (2023); Wang et al. (2021) can result in misalignment. Furthermore, the limited size of available medical data may prevent the full unleashing of the potential of LLMs.

To address these challenges, we introduce a novel concept: counterfactual explanations (CEs). The concept of CEs He et al. (2022); Virgolin & Fracaros (2023) has surfaced in machine learning as an insightful tool to comprehend how models' decisions can be changed. CEs offer a hypothetical alternative to the observed data, allowing for the assessment and comprehension of models through 'what if' scenarios. This technique has been integrated into diagnostic models to not only improve diagnostic accuracy but also enhance explainability Tanyel et al. (2023); Dai et al. (2022). For example, Tanyel *et al.* Tanyel et al. (2023) propose CEs to identify the minimal feature change and effectively demonstrate which features are more informative in differentiating two tumor types from MRI brains. Inspired by this, we aim to make this progress further interactive and explainable by explaining the global feature change in specific local regions. In particular, we propose CEs as *'what if we exchange the patch between two images, will the diagnosis shift?'* to identify critical regions within images that may cover abnormalities or lesions, providing insights into the diagnosis process. For instance, as illustrated in Figure.1, we generate a counterfactual image by iteratively swapping a patch between semantically similar images with different diagnosis labels until a shift in predicted diagnosis is achieved. Due to aforementioned similarities, exchanging a patch in the same position between two radiology images − particularly those that are semantically similar but carry different labeled diagnoses − does not disrupt the anatomical content. Notably, previous methods only integrate the CEs into the decision-making process, lacking the ability to convey factual or counterfactual information effectively. In contrast, we translate our CEs into a prompt that can present the key concept of CEs and encapsulate the factual and counterfactual content. This prompt can yield more comprehensive instructions to LLMs and facilitate the elicitation of their knowledge.

In this paper, we propose a **Co**unter**F**actual **E**xplanations-based framework (CoFE) for radiology report generation. CoFE is capable of learning non-spurious visual representations and effectively leverage the capabilities of LLMs. First, we introduce a novel type of CEs for RRG tasks and propose a counterfactual generation process through contrastive learning, which constructs a counterfactual image and a learnable prompt. Specifically, we adopt a negative sampling strategy to discover the most semantically similar negative sample from the data bank, based on text similarity and diagnosis label. By iteratively exchanging patches between factual (positive) and negative samples until a predicted diagnosis change occurs, we pinpoint the critical region and create a counterfactual image. We then employ contrastive learning within a joint optimization framework to differentiate representations between factual and counterfactual samples, enabling the model to learn non-spurious visual representations. Subsequently, we employ a pretrained LLM, GPT-2 Medium Radford et al. (2019), as a decoder to generate reports. To fine-tune the LLM efficiently, we propose a learnable prompt that encapsulates both factual and counterfactual content. This prompt can elicit the embedded knowledge within the LLM, which is helpful to generate semantically coherent and factually complete reports.

We evaluate our proposed method on two benchmarks, IU-Xray Demner-Fushman et al. (2016) and MIMIC-CXR Johnson et al. (2019) respectively. Extensive experiments demonstrate that our approach can outperform previous competitive methods in metrics that measure descriptive accuracy and clinical correctness. It indicates that leveraging CEs to learn non-spurious representations and prompt the generation process can improve the quality of predicted reports.

## 2 RELATED WORK

### 2.1 MEDICAL REPORT GENERATION

The pursuit of automating medical report generation through machine learning aims to alleviate the workload on radiologists. Numerous effective concepts exist to learn non-spurious representations to mitigate inherent data bias. Relation memory Chen et al. (2020; 2021) can prompt enhancement by boosting interaction efficiency of cross-modal memory network. Integrating medical knowledge is another solution, researchers utilize graph structural data Yang et al. (2022); Li et al. (2023) or medical tags Li et al. (2022b); Jing et al. (2018) to incorporate prior knowledge into image encoding. Additional models Yang et al. (2023a); Xu et al. (2023) also enhance performance by integrating knowledge information, with strategies including multi-modal semantic alignment and multi-label classification pre-training. To identify the abnormalities, PPKED Liu et al. (2021a) employs a unique architecture to mimic human learning processes. Tanida *et al.* Tanida et al. (2023) utilize a lesion detector pre-trained by pseudo labels to attain the non-spurious features and lead a pretrained GPT-2 Radford et al. (2019). Due to the lack of annotations, weakly supervised signals from pseudo labels may lead to the misalignment. Although, large pretrained models showcase the adaptability in learning medical representations Mohsan et al. (2023), the scarcity of data may limit the potential of LLMs. In this paper, our method concentrates on learning non-spurious representations by identifying critical regions, and employing a robust prompt to fine-tune the LLM efficiently.

### 2.2 COUNTERFACTUAL EXPLANATIONS REASONING

The advent of counterfactual explanations (CEs) construction has driven significant innovations, particularly in computer vision applications, enhancing both accuracy and interpretability. CEs have the potential to relieve existing methodologies from the reliance on extensive training data and meticulous annotations, by asking 'what if' scenario to explore self-supervised signals. Fang *et al.* Fang et al. (2019) and Kim *et al.* Kim et al. (2021) have introduced systems and frameworks, such as Counterfactual Generative Networks (CGN), designed to augment interpretability and resilience to CEs inputs without compromising accuracy. Similarly, CPL He et al. (2022) proficiently generates counterfactual features and has exhibited remarkable efficacy in tasks like image-text matching and visual question answering. Ji *et al.* Ji et al. (2023) specifically target video relationship detection, constructing CEs to elucidate their influence on factual scenario predictions. Further, the studies by Yang *et al.* Yang et al. (2023c) on PubMedQA highlight the crucial role of CEs, generated via ChatGPT, in learning causal features in counterfactual classifiers, demonstrating the versatility and broad applicability of counterfactual methods across various domains. In this paper, we employ CEs to enhance the RRG models, especially where acquiring a substantial amount of golden annotations is prohibitively expensive.

### 2.3 PROMPT TUNING

Prompt tuning is a method in natural language processing (NLP) used to efficiently modify the behavior of a pre-trained LLM, based on specific prompts or trigger phrases. This approach involves fine-tuning the model on a set of prompts or queries and their corresponding responses, allowing the model to respond more accurately or appropriately to those or similar prompts Shin et al. (2020); Zhou et al. (2022b). For example, Guo *et al.*Guo et al. (2022) applied Q-Learning to optimize soft prompts, while PTuning v2 Liu et al. (2021c) demonstrated that continuous prompt tuning could match the performance of fine-tuning in various scenarios. This technique has also garnered significant interest in the field of computer vision. CoOp Peng et al. (2021) introduced a strategy for continuous prompt optimization to negate the need for prompt design, and CoCoOp Zhou et al. (2022a) expanded upon this by learning an instance-conditional network to generate a unique input-conditional token for each image. Fischer *et al.* Fischer et al. (2022) also prove the adaptability

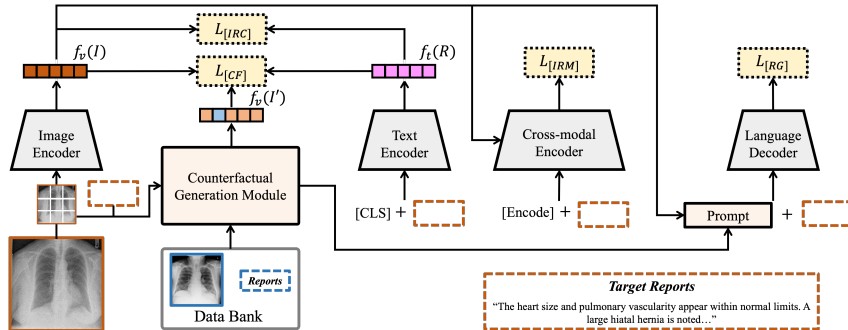

Figure 2: Illustration of our proposed **Co**unter**F**actual **E**xplanations-based framework (CoFE). CoFE consists of two unimodal encoders, one cross-modal encoder, one language decoder and our proposed counterfactual generation module that can construct a counterfactual image and a learnable prompt, respectively. The entire framework is trained through joint optimization, mainly employing contrastive learning paradigms for radiology report generation.

of prompt tuning in medical image segmentation tasks. However, the reliance on empirical risk minimization presents challenges, necessitating advancements to avoid spurious representations. In this paper, we aim to propose a generalizable prompt incorporating factual and counterfactual content to efficiently refine the medical LLMs.

## 3 METHODOLOGY

In this section, we introduce the detailed implementations of our proposed **Co**unter**F**actual **E**xplanations-based framework (CoFE). As shown in Fig.2, our CoFE mainly consists of two unimodal encoders, one cross-modal encoder, a language decoder, and a counterfactual generation module with four training objectives. We first introduce the backbone of CoFE and then describe the counterfactual generation process in detail.

### 3.1 BACKBONE

**Notations.** In this work, we aim to integrate counterfactual explanations into report generation models to learn non-spurious visual representations and efficiently generate high-quality reports. Radiology report generation tasks require a model to translate a complex radiology image $I$ into a generic report $T = \{y_1, y_2, \ldots, y_n\}$. We denote the target report by $\hat{T} = \{\hat{y}_1, \hat{y}_2, \ldots, \hat{y}_{\hat{n}}\}$. $n$ and $\hat{n}$ represent the number of tokens in a report. In addition to corresponding reports, we also utilize the diagnosis label $C$ for each examination in this work. Since not all existing benchmarks provide such annotations, we use the CheXPert labeling tool Irvin et al. (2019) to label ground truth reports with 14 different medical terminologies. Notably, we assign the label "No Finding" when CheXPert does not extract any terminologies.

Automatic report generation systems are typically based on encoder-decoder frameworks. The encoder generally aims to convert the given image $I$ into dense visual vectors $f_v(I)$. The decoder is usually a sequence processing network, which translates $f_v(I)$ to a report $T$. In this work, we adopt the BLIP Li et al. (2022a)-based architecture as the backbone to generate desired generic and matched reports, drawing inspiration from the successful concepts found in DCL Li et al. (2023). Such architecture presents superior representation learning capabilities and employs three losses to train two uni-modal encoders, one cross-modal encoder, and one language decoder.

**Image encoder**. Different from prior work employing CNNs, we use a pre-trained ViT Dosovitskiy et al. (2020)-S as the image encoder $f_v(\cdot)$. ViT enables finer semantic feature extraction by dividing images into more patches, specifically 16×16, compared to conventional CNNs 7×7. A `[CLS]` token is also prepended before input to the encoder layers. The encoder layer process, $\mathbf{f}_e(\cdot)$, is defined as:

$$\mathbf{f}_e(x) = \text{LN}(\text{FFN}(e_{attn}) + e_{attn}), \qquad (1)$$

$$e_{attn} = \text{LN}(\text{MHA}(x) + x), \qquad (2)$$

where FFN and LN represent Feed Forward Network Vaswani et al. (2017) and Layer Normalization operation Ba et al. (2016), respectively. MHA Vaswani et al. (2017) (multi-head attention) splits attention into $n$ heads, with each head, $\text{Att}(\cdot)$, defined as:

$$\text{Att}(x) = \text{softmax}(\frac{\mathbf{Q}^x(\mathbf{K}^x)^\top}{\sqrt{d}})\mathbf{V}^x. \tag{3}$$

with $d = 384$ being the embedding space dimension, and $\{\mathbf{Q}, \mathbf{K}^*, \mathbf{V}^*\}$ representing the corresponding *Query, Key, Value* vectors. The resulting output, the encoded visual vectors $\mathbf{f}_I$, will be used for report generation.

**Text encoder**. We employ a PubMedBERT Gu et al. (2020), pre-trained with abstracts and full texts from PubMed[1], as our text encoder $f_t(\cdot)$. It extracts textual representations $f_t(T)$ from positive and negative reports, which will be utilized for calculating the image-report contrastive (IRC) loss to facilitate learning robust and generalizable medical visual and textual representations. We utilize momentum image and text encoders to extract positive and negative data representations in a batch. Then we first calculate the softmax-normalized image-to-report similarity $f_m^{\text{i2t}}(I)$ and the report-to-image similarity $f_m^{\text{t2i}}(T)$ for the image $I$ and its paired report $T$ by $f_m^{i2r}(I) = \frac{\exp s(I,T_m)/\tau}{\sum_{m=1}^M \exp s(I,T_m)/\tau}$, with $\tau$ as a learnable temperature parameter. The IRC loss can be written as:

$$\mathcal{L}_{\text{IRC}} = \frac{1}{2}(\mathcal{L}_{\text{ce}}(g^{t2i}(T), f^{t2i}(T)) + \mathcal{L}_{\text{ce}}(g^{i2t}(I), f^{i2t}(I))). \tag{4}$$

where $g(\cdot)$ denotes the ground truth of similarity.

**Cross-modal encoder**. The cross-modal encoder is utilized to capture the cross-modal representations given an image-report pair, which contains multiple Transformer sub-modules. Each sub-module is composed of a bidirectional self-attention layer, a cross-attention layer and a feed-forward neural network composition. In cross-attention layer, for each head, $\{\mathbf{Q}, \mathbf{K}^*, \mathbf{V}^*\}$ comes from $\mathbf{Q} = W_q * e_{attn}$, $\mathbf{K} = W_k * \mathbf{f}_I$, and $\mathbf{V} = W_v * \mathbf{f}_I$, where $W_*$ are the learnable parameters. Then [Encode] vector is projected to $d = 2$ with a linear layer to predict the probability $p^{itm}$. Then the image-report matching (IRM) loss is conducted as following to identify whether the given image-report pair is positive (matched) or negative (unmatched):

$$\mathcal{L}_{\text{IRM}} = \mathcal{L}_{\text{ce}}(g^{irm}, p^{irm}). \tag{5}$$

**Language decoder**. Acknowledging the superior capabilities of LLMs in various language generation tasks, we employ a GPT-2 Radford et al. (2019) Medium, also pre-trained from PubMed, as our language decoder. This enables the generation of detailed and semantically coherent reports. GPT-2, an auto-regressive model leveraging self-attention, conditions each output token in a sequence on its previous tokens for report generation. The entire process can be represented as:

$$p(T|I) = \prod_{t=1}^n p(y_t|y_1, \ldots, y_{t-1}, I). \tag{6}$$

Here, $y_t$ is the input token at time step $t$. The typical objective for report generation is minimizing cross-entropy loss between the predicted and ground truth token sequences. With ground truth report $\hat{R}$, all modules are optimized to maximize $p(\mathbf{y}|I)$ by minimizing:

$$\mathcal{L}_{\text{RG}} = -\sum_{t=1}^{\hat{n}} \log p(\hat{y}_t|\hat{y}_1, \cdots, \hat{y}_{t-1}, I). \tag{7}$$

### 3.2 COUNTERFACTUAL GENERATION

In this section, we will explain how to generate counterfactual features, encompassing a counterfactual image and a learnable prompt in detail. Counterfactual images are pivotal, allowing the model to discern non-spurious features through contrasting representations between factual and counterfactual images. The learnable prompt encapsulating both factual and counterfactual contents then efficiently refine the pre-trained LLM.

---

[1] https://pubmed.ncbi.nlm.nih.gov

**Negative sampling strategy**. Counterfactual features combine features derived from both factual (positive) and negative data. We first propose a negative sampling strategy to select the negative data from a data bank. Such negative data should have different labels and are difficult to be distinguished from the factual data. To implement this, we first construct a data bank, denoted by $D$, containing candidate data, each instance $d_i$ is annotated with {Image $I_i$, Report $T_i$, Label $C_i$}, maintaining balanced distribution of diagnostic labels within the data bank. Next, we select the negative data from the data bank, as $d^- = \arg\max_i BLEUScore(T, T_i)$ and $C \neq C_i$. The BLEUScore function calculates the BLEU Papineni et al. (2002) score, setting the factual report as reference and the negative data as candidate. The so-selected $d^- = I^-, T^-, C^-$ is earmarked as negative data, exhibiting textual semantics similar to the original data but possessing distinct labels, emphasizing their inherent dissimilarity. The entire procedure is visually depicted in Fig. 3.

**Counterfactual image**. After selecting the negative data from candidates, we proceed to generate counterfactual images combing factual and negative images, thereby enhancing non-spurious representations through contrastive learning. As shown in Fig.4, a factual image, $I$, is presented in the form of $n$ patches: $I = p_1, p_2, ..., p_n$; its corresponding negative image is represented as $I^- = p_1^-, p_2^-, ..., p_n^-$. From the 1st to the $n$-th patch, each patch of the negative image replaces the patch of the factual image at the corresponding position. The modified image is denoted by $I' = (1 - u) * I + u * I^-$, where $u$ is a one-hot vector to present the index of the replaced patch. Subsequent to each replacement, the modified image is fed into a pre-trained and frozen discriminator composed of the image encoder and a Multilayer Perceptron (MLP)

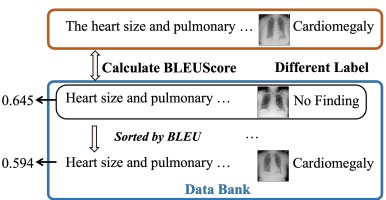

Figure 3: Illustration of negative sampling strategy. The objective is to select a negative sample that is mostly similar in semantics but carries a different diagnostic label from the data bank.

to predict the logits for the diagnostic label $C'$. The replacement process ceases once $C' \neq C$, culminating in the acquisition of the counterfactual image $I'$. This methodology enables the identification of critical regions that prompt models to alter the predicted diagnosis. In essence, such regions contain pivotal information pertinent to the examination. It helps to mitigate inherent data bias and facilitate the model's focus on these critical regions, learning non-spurious and robust visual representations.

**Learnable prompt**. Another key component of our counterfactual features is a learnable prompt, designed to elicit knowledge and leash the potential of pre-trained LLMs. Frequently used prompts in caption tasks, such as "the caption is..." or "describe [visual tokens]", clarify the task but often lack comprehensive instructions. To rectify this deficiency, we embed both factual and counterfactual content within the learnable prompt to attain more generalizable representations. As suggested by Tu et al. (2023), our prompt incorporates detailed instructions and is formulated by concatenating the factual visual tokens, factual label, counterfactual label, and the index of the patch with supplementary text. The training prompt is articulated as "Replacing the $u$ patch of $f_v(I)$ can lead to a shift in the predicted diagnosis from $C$ to $C^-$. The diagnostic report, describing critical entities including tubes, pneumothorax, pleural effusion, lung opacity, cardiac silhouette, hilar enlargement, and mediastinum, is".

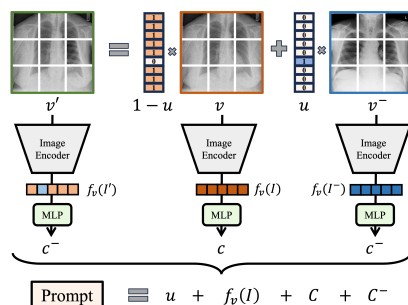

Figure 4: Illustration of counterfactual generation process, including a counterfactual image and a learnable prompt.

### 3.3 JOINT OPTIMIZATION

In addition to the image-report matching loss, image-report contrastive loss, and report generation loss, we introduce a novel contrastive loss aimed at amplifying the proficiency of visual representation learning. Specifically, the factual image feature $f_v(I)$, text feature $f_t(T)$, and counterfactual image feature $f_v(I')$ are employed to compute the counterfactual loss, $\mathbf{L}_{cf}$, thereby extending the divergence between the counterfactual features and the features of the original data. This can be represented as:

$$\mathcal{L}_{CF} = -\log \frac{e^{\frac{f_v(I)@f_t(T)}{\tau}}}{e^{\frac{f_v(I)@f_t(T)}{\tau}} + e^{\frac{f_v(I')@f_t(T)}{\tau}}} \tag{8}$$

Here, @ denotes cosine similarity. The total training loss is written as:

$$\mathcal{L} = \mathcal{L}_{IRC} + \lambda rg\mathcal{L}_{RG} + \mathcal{L}_{IRM} + \lambda_{cf}\mathcal{L}_{CF} \tag{9}$$

where $\lambda_{cf}$ and $\lambda_{rg}$ denote the loss weights, we assign a value of 2 to $\lambda_{cf}$ and 5 to $\lambda_{rg}$ based on performance on the validation set.

# 4 EXPERIMENTS

## 4.1 DATASETS, EVALUATION METRICS AND SETTINGS

**Datasets**. We validate the efficacy of our proposed CoFE using the **IU-Xray** Demner-Fushman et al. (2016) and **MIMIC-CXR** Johnson et al. (2019) benchmarks. The settings adopted by Chen et al. (2020) are utilized to uniformly split and preprocess the datasets and reports, ensuring a fair comparison. IU-XrayDemner-Fushman et al. (2016), a prevalent benchmark for evaluating RRG systems, comprises 3,955 reports and 7,470 images. After excluding cases with only one image as per Chen et al. (2020); Li et al. (2019), 2069/296/590 cases are allocated for training/validation/testing respectively. We utilize CheXPert to extract terminologies from reports and assign labels to each examination. MIMIC-CXRJohnson et al. (2019), the most extensive radiology dataset publicly-available, includes 368,960 images and 222,758 reports. It has officially segmented subsets and has spurred the development of structurally explorative child datasets like RadGraph Jain et al. (2021).

**Metrics**. We employ two types of metrics to evaluate the quality of our predicted reports. First, **natural language generation** (NLG) are employed to assess the descriptive precision of the predicted reports, with CIDEr Vedantam et al. (2015) and BLEU Papineni et al. (2002) being primary. BLEU is primarily designed for machine translation, evaluating word n-gram overlap between reference and candidate, repeating frequent sentences can also achieve high scores. Conversely, CIDEr, developed for captioning systems, rewards topic terms and penalizes frequent ones, thus is more fitting for evaluating reports in RRG tasks. Additionally, ROUGE-L Lin (2004) and METEOR Banerjee & Lavie (2005) are also considered for comprehensive comparison. Lastly, **clinical efficacy** metrics, a more recent innovation, ascertain the clinical accuracy of reports by using the CheXPert labeling tool to annotate predicted reports. Subsequent classification measurements like F1-Score, Precision, and Recall assess the aptness of the generated reports in describing abnormalities.

**Experimental settings.** For both datasets, we only utilize the front view examinations. We first pretrain the ViT-S for 10 epochs using diagnosis labels. Given the distinct domain difference between medical and general texts, a pretrained PubMedBert Gu et al. (2020) is utilized as both a tokenizer and a text encoder. The training is conducted on 4 NVIDIA 2080 Ti GPUs, spanning 50 epochs with batch sizes of 8. The model checkpoint achieving the highest CIEDr metric is selected for testing. An Adam optimizer, with a learning rate of 1e-4 and a weight decay of 0.02, is applied. We set the size of data bank to 1,380. Note that all encoded vectors are projected by a linear transformation layer into a dimension of $d = 384$.

## 4.2 MAIN RESULTS

**Descriptive Accuracy.** We compare our CoFE with several competitive RRG methods on two benchmarks. R2Gen Chen et al. (2020) and CMN Chen et al. (2021) are two widely-used baseline models implementing relation memory. KERP Li et al. (2019), PPKED Liu et al. (2021a), MKG Zhang et al. (2020) and MGSK Yang et al. (2022) are proposed to integrate medical knowledge with typical RRG backbones. CMCL Liu et al. (2022) and DCL Li et al. (2023) employ contrastive learning to further improve performance. As presented in Table.1, our method notably outperforms all competing approaches, attaining the highest figures across almost all the metrics, with a CIDEr score of 0.766 and BLEU-4 score of 0.170 on IU-xray. Similarly, our method demonstrates competitive performance on the MIMIC-CXR dataset, achieving the highest ROUGE-L score

Table 1: The performance in NLG metrics of our proposed method compared to other competitive methods on the IU-Xray and MIMIC-CXR datasets. The highest figures in each column are highlighted in bold.

| IU-Xray | | | | | MIMIC-CXR | | | | |
|---|---|---|---|---|---|---|---|---|---|
| Methods | CIDEr | BLEU-4 | ROUGE-L | METEOR | Methods | CIDEr | BLEU-4 | ROUGE-L | METEOR |
| R2Gen | 0.398 | 0.165 | 0.371 | 0.187 | R2Gen | 0.253 | 0.103 | 0.277 | 0.142 |
| KERP | 0.280 | 0.162 | 0.339 | - | CMN | - | 0.106 | 0.278 | 0.142 |
| HRGP | 0.343 | 0.151 | 0.322 | - | TopDown | 0.073 | 0.092 | 0.267 | 0.129 |
| MKG | 0.304 | 0.147 | 0.367 | - | PPKED | 0.237 | 0.106 | 0.284 | 0.149 |
| PPKED | 0.351 | 0.168 | 0.376 | 0.190 | RGRG | **0.495** | **0.126** | 0.264 | 0.168 |
| MGSK | 0.382 | **0.178** | 0.381 | - | MGSK | 0.203 | 0.115 | 0.284 | - |
| CMCL | - | 0.162 | 0.378 | 0.186 | CMCL | - | 0.097 | 0.281 | 0.133 |
| DCL | 0.586 | 0.163 | 0.383 | 0.193 | DCL | 0.281 | 0.109 | 0.284 | 0.150 |
| **CoFE** | **0.766** | 0.170 | **0.524** | **0.206** | **CoFE** | 0.454 | 0.121 | **0.296** | **0.171** |

of 0.296 and METEOR score of 0.171. These results showcase the superior capability of our method in generating matched and semantically similar reports.

**Clinical Correctness.** We also evaluate our method by Clinical Efficacy (CE) metrics on the MIMIC-CXR dataset to evaluate the clinical correctness of our predicted reports. In Table. 2, we compare the performance against several baseline models, DCL, R2Gen and MKSG, respectively. Most notably, our CoFE achieves the SOTA performance across all the clinical efficacy metrics, with a Precision of 0.486, Recall of 0.369, and F1-score of 0.402. This performance-boosting underscores effectiveness of integrating counterfactual explanations, enabling the model to generate more clinically correct and relevant reports.

## 4.3 ANALYSIS

In this section, we conduct ablation studies and a case study on IU-Xray and MIMIC-CXR datasets to investigate the proficiency of each key component in CoFE. Specifically, Table. 3 presents the quantitative analysis of CoFE on IU-Xray measuring descriptive accuracy. And clinical correctness evaluation is reported in Table. 2. We employ a vanilla BLIP as our base model.

**Effect of pre-trained LLMs**. Compared with the base model in setting (a), illustrated in Table 3, where we utilize a pre-trained PubMedBert and a 355M-parameter GPT-2 Meduium as the text encoder and language decoder, there is a significant enhancement in all metrics, with CIDEr improving from 0.366 to 0.517, emphasizing the impactful role of LLMs in enhancing the report generation performance. Specifically, PubMedBert can encode the reports into better textual representations, while GPT-2 has the capability to generate semantically-coherent and logically-consistent reports.

**Non-spurious Representation Learning**. The primary motivation for integrating counterfactual explanations is to enhance non-spurious visual representations by contrasting the representations between factual and counterfactual images. When comparing setting (c) to Setting (a) and the full model to setting (b), a significant performance boost is observable across all metrics. For instance, CIDEr elevates from 0.517 to 0.680 and from 0.706 to 0.766, respectively. Additionally, BLEU-4 metrics reach 0.170, achieving the SOTA performances. These notable elevations highlight the importance of non-spurious representation learning capabilities in radiology report generation tasks.

Table 2: The comparison of the clinical efficacy metrics on MIMIC-CXR dataset.

| Methods | Precision | Recall | F1-score |
|---|---|---|---|
| DCL | 0.471 | 0.352 | 0.373 |
| R2Gen | 0.333 | 0.273 | 0.276 |
| MKSG | 0.458 | 0.348 | 0.371 |
| Base | 0.328 | 0.275 | 0.279 |
| + LLMs | 0.394 | 0.321 | 0.314 |
| + prompt | 0.462 | 0.350 | 0.364 |
| + $\mathcal{L}_{CF}$ (full) | **0.486** | **0.369** | **0.402** |

**Effect of Prompt Tuning**. To fully elicit pre-trained knowledge and unleash the potential of LLMs, we propose a learnable prompt that encapsulates both factual and counterfactual content to refine the LLMs. Observing setting (a) *vs* (b) and (c) *vs* the full model, it is evident that our proposed prompt can further augment performance, especially evident

Table 3: Quantitative analysis of our proposed method on the IU-Xray dataset. We employ a vanilla BLIP without loading pre-trained parameters as the base model.

| Settings | LLMs | Prompt | $\mathcal{L}_{CF}$ | Random Sampling | CIDEr | BLEU-4 | ROUGE-L | METEOR |
|----------|------|--------|------|-----------------|-------|--------|---------|--------|
| Base | | | | | 0.366 | 0.130 | 0.259 | 0.157 |
| (a) | ✓ | | | | 0.517 | 0.148 | 0.392 | 0.184 |
| (b) | ✓ | ✓ | | | 0.706 | 0.157 | 0.472 | 0.189 |
| (c) | ✓ | | ✓ | | 0.680 | 0.155 | 0.490 | 0.198 |
| (d) | ✓ | | ✓ | ✓ | 0.659 | 0.158 | 0.474 | 0.190 |
| (e) | ✓ | ✓ | ✓ | ✓ | 0.702 | 0.163 | 0.481 | 0.204 |
| CoFE | ✓ | ✓ | ✓ | | **0.766** | **0.170** | **0.524** | **0.206** |

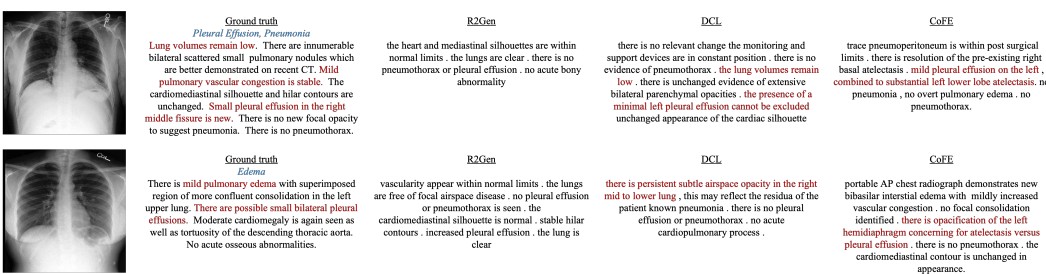

Figure 5: Illustration of reports generated by R2Gen, DCL and our CoFE. The text in blue demonstrates the ground truth diagnosis labels. The red text represent the accurately matched abnormalities.

in ROUGE-L, which elevates from 0.392 to 0.472 and from 0.490 to 0.524, respectively. This increment underscores the effectiveness of our prompt in refining the model's natural language generation capability. Furthermore, as shown in Table.2, this prompt can also increase the clinical correctness of the predicted reports.

**Negative Sampling Strategy**. The key point to construct counterfactual image is selecting negative data which have different labels and are difficult to be distinguished from the factual data. To verify this, we employ a random sampling strategy in which candidate data are indiscriminately selected as the negative sample. The incorporation of this random sampling strategy in settings (d) and (e) results in a discernible degeneration in the model's capability to generate high-quality reports. This slight decline across almost all performance metrics elucidates the influential role of our negative sampling strategy in pinpointing the most suitable negative data.

**A case study**. In Figure.5, we present two samples from MIMIC-CXR and corresponding reports generated by R2Gen, DCL and our CoFE. R2Gen seems to lack specificity and detailed insights, providing a more generalized statement about the conditions and missing several key abnormalities mentioned in the ground truth, such as pulmonary nodules and pleural effusion. The DCL model is somewhat more aligned with the ground truth, acknowledging the unchanged appearance of the cardiac silhouette and the presence of extensive bilateral parenchymal opacities. However, it fails to mention the presence of pulmonary nodules and the pleural effusion in the right middle fissure specifically. In contrast, CoFE addresses pleural effusion, atelectasis, and the absence of pneumonia and pneumothorax, making it more in alignment with certain elements of the ground truth. These observations prove that our CoFE is capable of generating factual complete and consistent reports.

## 5 CONCLUSION

In this paper, we present a novel framework, Counterfactual Explanations-based Framework (CoFE), designed for radiology report generation. To address the inherent data bias, we introduce a novel counterfactual concept, allowing CoFE to identify critical regions and construct a counterfactual image during training. By contrasting the representations between factual and counterfactual features, CoFE is adept at learning non-spurious visual representations. Subsequently, we summarize the counterfactual generation process into a learnable prompt, enabling the efficient fine-tuning of a pre-trained LLM. Experiments on two widely-recognized benchmarks verify the efficacy of our approach in generating factual, comprehensive, and coherent reports.

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
