# OpenReview forum: "Learning with Counterfactual Explanations for Radiology Report Generation"
_ICLR.cc/2024/Conference — ICLR 2024 Conference Withdrawn Submission_

### Official Review · Reviewer_nNsA · 2023-10-24

**Soundness:** 2 fair
**Presentation:** 3 good
**Contribution:** 2 fair
**Rating:** 5
**Confidence:** 5

**Summary:**

This work experiments with generating counterfactual images by overlapping crops of different pathologies and using them during training along with counterfactual text prompts that are created based on image and counterfactual labels.

**Strengths:**

The paper takes an interesting approach to incorporate counterfactual images into training.

**Weaknesses:**

The approach presented by the paper is interesting. My primary concern is that the complexity is high and not clearly justified.

Why not just simplify the process so that the input samples just contained the synthetic counterfactuals instead of having multiple losses?

Why not demonstrate improved performance on a classification task instead of a complicated language generation task where the evaluation metrics are more complicated and imprecise?

The paper claims:
> By contrasting the representations between factual and counterfactual features, CoFE is adept at learning non-spurious visual representations

which I cannot find evidence to support in the paper. Better language generation by itself does not allow us to conclude that no spurious features are being used. A controlled experiment that would allow the direct conclusion of features used, and specifically avoided, would be better at supporting this claim.

**Questions:**

Questions mentioned in weaknesses.

---

### Official Review · Reviewer_QvNC · 2023-10-30

**Soundness:** 3 good
**Presentation:** 3 good
**Contribution:** 2 fair
**Rating:** 5
**Confidence:** 3

**Summary:**

The paper proposes a novel framework for the counterfactual explanation of radiology report synthesis. The authors propose using a negative sampling strategy to discover counterfactual patches that affect the synthesis. Then the framework uses contrastive learning to distinguish factual and counterfactual representations. Finally, the GPT-2 is instructed with learnable prompts for report synthesis.

**Strengths:**

1. The paper is well-written with a clear storyline that motivates the framework.
2. It is a novel concept to involve multimodal information in the counterfactual explanation of radiology.
3. The evaluation of counterfactual effectiveness is comprehensive and detailed.

**Weaknesses:**

1. The paper should argue the validity of GPT-2, and the reason why more advanced (and publicly available) LLMs like LLaMA and GPT-3 are not used. Also, it will be beneficial if the authors discuss the feasibility of using the in-context capability of LLM to enhance the pipeline.

2. The effectiveness and efficiency of the negative sampling strategy are not assured. Authors should consider using more effective approaches to search/optimize counterfactual images. Recent years have witnessed the emergence of generative model-based outlier synthesis [1,2,3,4]. The paper shall discuss the feasibility or infeasibility of adopting them to obtain the counterfactual images.

3. When swapping the patches, how to maintain the ground truth (e.g., semantic contours, informative attributes) of the target image remains a challenge. The paper should discuss in more details how well the the ground truth is preserved.

[1] SemanticAdv: Generating Adversarial Examples via Attribute-conditional Image Editing. (ECCV 2020)

[2] Zero-Shot Model Diagnosis. (CVPR 2023)

[3] Adversarial Counterfactual Visual Explanations. (CVPR 2023)

[4] LANCE: Stress-testing Visual Models by Generating Language-guided Counterfactual Images.

**Questions:**

1. Please address my concerns stated in the weakness section. Considering the current status of the paper, I will rate it as a borderline reject. However, I look forward to the authors' response and I will consider revising the rating based on the soundness of the response.

---

### Official Review · Reviewer_u7dz · 2023-10-31

**Soundness:** 3 good
**Presentation:** 2 fair
**Contribution:** 3 good
**Rating:** 5
**Confidence:** 4

**Summary:**

Addressing the challenge of limited explainability in diagnostic report generation models, the authors have introduced the CounterFactual Explanations-based framework (CoFE) tailored for radiology report generation. This approach significantly enhances the model’s capability to answer 'what if' questions, enabling it to learn more authentic and reliable visual representations by contrasting factual and counterfactual images. The efficacy of CoFE has been demonstrated through experiments on two benchmark datasets, where it consistently generated more stable and reliable results, showcasing the value of counterfactual explanations in improving model performance and interpretability.

**Strengths:**

+ The authors have innovatively applied counterfactual explanations to Radiology Report Generation, introducing a novel framework along with a counterfactual generation process via contrastive learning. They adopted a sophisticated negative sampling strategy, ensuring the selection of semantically similar negative samples from the database for enhanced contrastive learning.
+ The decision to fine-tune a pre-trained language model, specifically GPT-2 Medium, for the final report generation demonstrates strategic thinking. The introduction of prompts is another commendable approach, aiding in the refinement of model weights for superior performance.

**Weaknesses:**

- The generation method for counterfactual images appears to be limited, assuming that a single image patch can result in a counterfactual outcome. This approach may not capture scenarios where multiple patches collectively contribute to a counterfactual result, which could be a critical aspect in understanding complex medical images.

- Addressing multiple patches simultaneously would lead to a computational complexity of 2n2n, which is not practical. The authors should acknowledge this limitation and discuss potential ways to manage this complexity, or propose alternative strategies to handle multiple patches.

- The use of prompts in the methodology is unclear and seems inconsistent. Typically, prompts should be uniform across tasks to ensure consistency. However, Figure 4 suggests that the prompt includes image-specific information, which could create confusion. The authors need to clarify their use of prompts and maintain a consistent approach throughout.

- Section 3.1 of the methodology, which covers existing concepts and methods, seems misplaced. It would be more appropriate to move this content to the related work section, allowing Section 3.2 to solely focus on the novel aspects of the proposed framework. This would streamline the paper and highlight the unique contributions of the work.

-  To convincingly demonstrate the efficacy of the proposed method, the authors should present results based on the best counterfactual images generated at the end of training. This would provide tangible evidence of performance improvement and validate the approach.

**Questions:**

- The authors need to provide a clear explanation of what the modified images represent, especially if they are a linear combination of two images, and whether this could lead to nonsensical images.
- Clarification is needed on whether all parameters of the pre-trained language model were fine-tuned during training or if some were frozen, as this detail is crucial for understanding the experimental setup and potential implications on model performance.

---

### Official Review · Reviewer_sgJL · 2023-10-31

**Soundness:** 2 fair
**Presentation:** 2 fair
**Contribution:** 1 poor
**Rating:** 3
**Confidence:** 4

**Summary:**

The authors propose generating counterfactual examples in order to fine-tune BLIP-style models. A counterexample for a given sample is obtained by searching for other instances with reports that are textually similar but have an image with different labels. The nearest such instance with a different label is then used to swap image patches between it and the original sample until the prediction shifts on the candidate. A downstream LLM is then fine-tuned by feeding in the encoded image, label, counterfactual label and location of the swapped patch as part of a detailed prompt. The authors find that the generated reports score better on a range of NLG metrics.

**Strengths:**

Radiology report generation is a difficult and important problem.

**Weaknesses:**

The key novelty of this paper seems to be generating counterfactuals as a way of generating more expressive prompts for fine-tuning. The latter itself is not new to the paper however. The results of this show some improvement in only some of the NLG metrics but there is not sufficient analysis as to why that is the case.

The authors claim that the way in which counterfactuals are generated does not “disrupt the anatomical content” but it strikes me as likely that the generated instances could in fact have disruptions due to anatomic misalignment, differences in contrast etc. The generated instances are therefore likely out-of-distribution for the image encoder so that there may be some doubts as to the validity of the discriminator network’s outputs, which uses the same image encoder.

**Questions:**

Can you offer some intuition as to why some NLG metrics improve but not others?
Can you clarify how you deal with instances with many labels?
Can you clarify your framework setup during inference?
How do you deal with images which may be semantically very similar but are shifted? Would your patch replacement strategy still work?